# A DOPO-Based Compound Containing Aminophenyl Silicone Oil for Reducing Fire Hazards of Polycarbonate

**DOI:** 10.3390/ma16041449

**Published:** 2023-02-09

**Authors:** Xiaoqing Song, Wendi Xu, Boyu Cai, Luze Wang, Zhonglin Luo, Biaobing Wang

**Affiliations:** Jiangsu Key Laboratory of Environmentally Friendly Polymeric Materials, School of Materials Science and Engineering, Jiangsu Collaborative Innovation Center of Photovoltaic Science and Engineering, Changzhou University, Changzhou 213164, China

**Keywords:** polycarbonate, P/N/Si-containing flame retardant, fire hazard, mechanical properties

## Abstract

A novel P/N/Si-containing flame retardant (marked as DASO) was synthesized through an Atherton–Todd reaction between 9,10-dihydro-9-oxa-10-phospha-phenanthrene-10-oxide and aminophenyl silicone oil, and further used for reducing fire hazards of polycarbonate (PC). The chemical structure of DASO was verified via FTIR, ^1^H, and ^31^P NMR. Upon the incorporation of 2 wt% DASO, the FRPC composite achieved a high limiting oxygen index (LOI) of 32.2% and a desired UL-94 V-0 rating. In this case, the peak heat release rate (PHRR) and total smoke production (TSP) were reduced by 26% and 44% as compared with the pure PC, respectively. The improved fire safety contributed to the flame retardant roles of DASO in both the condensed phase and gas phase. The presence of DASO promoted the formation of dense and highly graphited char layer in the condensed phase, and released non-combustible gases and phosphorus-containing radicals in the gas phase. Furthermore, the FRPC composites displayed comparable elongation at break but a slightly reduced tensile and impact strength.

## 1. Introduction

Bisphenol-A polycarbonate (PC) has been widely used in a variety of industrial fields owing to its excellent comprehensive performances, including impact strength, dimensional stability, heat resistance, and transparency [1,2]. Although the PC itself shows a UL-94 V-2 rating for its capability of self-carbonation to a certain degree, it is still far below the stringent requirements of a V-0 rating for harsh applications [3,4]. As we all know, fire accidents bring about a great threat to human security and the environment; it is thus urgent to enhance the fire resistance of PC to expand its application. Over the past decades, a majority of halogenated flame retardants (FRs), although characterized by low cost and better flame retardant effect, have been forbidden for the release of some toxic gases such as halogenated dioxins and dibenzofurans during the combustion. Recently, many efforts have been devoted to preparing halogen-free FRs in order to meet the harsh regulations regarding environmental protection and human health. 

Up to date, different types of halogen-free FRs suitable for PC have been reported, such as silicon- [5,6], phosphorus- [7,8], sulfur- [9,10], nitrogen- [11,12], and multi-elements-based compounds [13,14]. However, satisfactory flame retardant efficiency was achieved merely at its high loading level for most halogen-free compounds containing a single FR element, thus generally deteriorating the mechanical properties of the PC matrix. Due to the synergistic effect of the different FR elements, the compounds containing multi-elements display better efficiency for enhancing the fire safety of the PC matrix. Consequently, many works have focused on the synthesis of multi-in-one flame retardants, such as P/N [15,16], P/S [17,18], P/Si [19,20], P/S/N [21,22], etc. Chen et al. [23] prepared PC-based composites blended with phosphazene-triazine bi-group flame retardant (A3); the PC/A3 composites reached a V-0 rating and 29.3% of LOI value in the case of 13.5% A3 loading level. Yang et al. [24] synthesized a halogen-free FR containing sulfonamide (FRSN), the incorporation of only 0.08 wt% FRSN endowed PC with an LOI value of 33.7% and V-0 rating. According to the report of Hu et al. [25], a novel P/Si-containing FR (P(DOPO-VTES)) was synthesized and adopted to improve the flame retardancy of PC. A maximal LOI value of 32.8% was achieved for PC/5%P(DOPO-VTES) composite, and a V-0 rating was further obtained for PC/5%P(DOPO-VTES)/2 wt% montmorillonite composite. 

However, the introduction of additive FRs sometimes deteriorates the mechanical properties of the PC composite due to their poor compatibility, especially in the case of the usage of an anti-dripping agent such as talc and PTFE. Therefore, it is still a challenge to endow the PC matrix with better flame retardancy meanwhile keep almost the mechanical properties. The effect stratagem is to improve the compatibility of the FRs with PC or avoid the usage of an anti-dripping agent. It was well accepted that silicon-based FRs mainly decomposed to form silicon-containing fragments for improving the degree of graphitization of the condensed phase [26,27,28] and also produced inorganic SiO_2_ acting as a protective barrier to keep the internal substrate from further burning. In the present work, a P/N/Si-containing FR (DASO) was synthesized using DOPO and aminophenyl silicone oil as raw materials. Subsequently, the DASO acting as both flame retardant and an anti-dripping agent was blended with PC. As expected, the UL-94 V0 rating was obtained for PC/2 wt% DASO composite accompanied with comparable mechanical properties. Furthermore, the overall properties of the FRPC composites were investigated, and the flame retardant effect on the gas and condensed phase was disclosed.

## 2. Experimental

### 2.1. Synthesis of DASO

DASO was prepared through Atherton–Todd reaction between DOPO and APSO (Figure 1). To a 250 mL three-neck flask equipped with a magnetic stirrer and a reflux condenser, DOPO (0.54 g, 2.5 mmol), APSO (24 g, 2.5 mmol), CH_2_Cl_2_ (50 mL), and triethylamine (0.25 g, 2.5 mmol) were introduced. Following that, DOPO was dissolved, CCl_4_ (0.385 g, 2.5 mmol) was added dropwise under stirring in an ice bath. Subsequently, the mixture was heated to 40 °C and kept the temperature for 12 h. Afterward, triethylamine salt and CH_2_Cl_2_ were removed by vacuum filtration and rotary evaporation, respectively. The crude product was dissolved in dichloromethane and washed with deionized water several times. Finally, after the removal of dichloromethane by rotary evaporation and vacuum drying at 100 °C for 24 h, it gives a yellow oily liquid (named DASO) with a yield of 91%.

### 2.2. Preparation of FRPC Composites

Prior to blending, the PC was vacuum-dried at 100 °C for overnight. The predetermined PC and DASO (Table 1) were blended on an internal mixer (US-70c, Suyan Technology CO., Ltd., Changzhou, China) at 250 °C for 5 min with a rotor speed of 60 r/min. The prepared blends were marked FRPC-X, wherein X denotes the weight percentage of DASO.

## 3. Results and Discussion

### 3.1. Characterization of DASO

The FTIR spectra of APSO, DOPO, and DASO are present in Appendix A. As seen, both APSO and DASO display some typical absorption peaks as follows: 3065 cm^−1^ (υ_φ-H_), 2965 (υ_CH3_) and 2905 cm^−1^ (υ_CH2_), 1260 and 804 cm^−1^ (υ_Si-C_), 1093 and 1023 cm^−1^ (υ_Si-O-Si_). Moreover, the peak at 2437 cm^−1^ ascribed to υ_P-H_ [29] disappears, while some characteristic absorption peaks at 1214 (υ_P-O-C_) and 932 cm^−1^ (υ_P-N_) are easily visible in the spectrum of DASO. It indicates that the DASO is synthesized successfully. 

Regarding the ^1^H NMR (Appendix A), the signals at 7–8 ppm are assigned to protons (c + e + f + h) of the benzene ring and -NH, and the one at 5.21 ppm ascribes to the proton (g) of -NH connected with CH_2_. The protons (a and b) from CH_3_ appear at 0.05 and 0.56 ppm, and the signal at 2.46 ppm is ascribed to proton (d) from CH_2_. As shown in ^31^P NMR (Appendix A), the chemical shifts at 14.51 and 15.72 ppm for DOPO merge into a single signal at 7.06 ppm for DASO, which might be triggered by the steric hindrance of phosphaphenanthrene group [30]. 

As presented in the TGA and DTG curves of DASO (Appendix A), the DASO displays a three-step degradation behavior with T_5%_ (5% mass loss temperature) of 293 °C, indicating the DASO fully meets the processing temperature of PC. The first stage originates from the breakage of P-N and P-C bonds, the second one with T_max_ (maximum decomposition temperature) of 467 °C is attributed to the breakage of the N-H and C-N bonds, and the last one with T_max_ of 580 °C includes the rapture of Si-C and Si-O-Si in the main chain.

### 3.2. Thermal Stability of PC and FRPCs

The thermal stability of the samples was assessed in terms of TGA, and the relative TGA and DTG curves are depicted in Figure 1, and Table 2 lists the values of T_5%_, T_max_, R_max_ (maximum decomposition rate), and CY_800_ (char yields at 800 °C). It is found that both PC and FRCPs display a one-step thermal decomposition process under a nitrogen atmosphere, revealing that the incorporation of DASO exerts no obvious changes on the thermal decomposition process of PC. However, the FRPCs present lower values of T_5%_ and T_max_ than the pure PC, which might be attributed to the fact that the DASO with less stable P-C and P-N bonds has a catalytic effect on the degradation of PC. Moreover, the introduction of DASO results in a reduced R_max_ but an enhanced CY_800_. For instance, the R_max_ is reduced from 33.4%/min for pure PC to 15.5%/min for FRPC-2; conversely, the CY_800_ is increased from 18.4 to 19.6%. It shows that the presence of DASO can catalyze the char-forming to impede the further thermal degradation of the PC matrix, thus exerting FR’s role on the condensed phase. Unlike a single-stage decomposition under a nitrogen atmosphere, three-stage decomposition behaviors occur under an air atmosphere. It indicates that the thermal degradation kinetics become much more complicated due to thermal oxidization. 

### 3.3. Flame Retardancy Analysis

Figure 2 illustrates the UL-94 rating and LOI values of the samples. As expected, flame retardancy improves steadily upon the increasing DASO contents. For example, the LOI value is increased from 25% for pure PC with a V-2 rating to 32.2% for FRPC-2 with a V-0 rating. Moreover, few char residues are produced during the LOI testing of pure PC; in contrast, much more compact char residues that can withstand the pressure of 20 g poise are left upon the incorporation of DASO.

### 3.4. Cone Calorimeter Tests

Figure 3 shows the curves of heat release rate (HRR), total heat release (THR), smoke production rate (SPR), and total smoke production (TSP), and Table 3 summarizes some specific parameters.

As seen, the time to ignition (TTI) shortens steadily with the increasing DASO content, for example, 150 s for pure PC > 130 s for FRPC-1 > 127 s for FRPC-2. The shortened TTI is mainly due to the earlier decomposition of DASO.

Generally, the THR and peak of HRR (PHRR) are valued as crucial parameters to reflect the combustion intensity. In comparison with THR of 74.2 MJ/m^2^ and PHRR of 443.5 kW/m^2^ for pure PC, these values reduce steadily with the incorporation of DASO. Especially, the minimum values, including 70.2 MJ/m^2^ of THR and 326.1 kW/m^2^ of PHRR, are achieved for FRPC-2. It demonstrates that the presence of DASO plays inhibiting function on the heat release during combustion. Likewise, toxic smoke is also considered a harmful factor in a real fire [31]. As shown in Figure 3c and Table 3, both SPR and TSP values reduce obviously with the loading of DASO. Particularly, FRPC-2 composite gives the lowest TSP of 35.4 m^2^ and PSPR (peak of SPR) of 0.15 m^2^/s, with a reduction of 44.2% and 55.9% by comparison with pristine PC, respectively. It reveals that DASO also has better smoke suppression. Additionally, the fire growth rate index (FIGRA = PHRR/T_PHRR_) is adopted to assess underlying fire spread or growth hazards during burning. As seen in Table 3, the FRPC-2 displays the lowest FIGRA value (2.04 kW/m^2^·s) among all samples, reflecting that fire safety is improved with the introduction of DASO.

The value of av-EHC (HRR/MLR) is often used to characterize the burning degree of volatiles in the gas phase. As seen, the av-EHC values offer a downward trend in the following order of PC (24.9 MJ/kg) > FRPC-1 (22.8 MJ/kg) > FRPC-2 (20.4 MJ/kg), demonstrating that DASO decreases the flammability of PC markedly. Additionally, PC/DASO composites give greater av-COY/av-CO_2_Y values (0.069 for FRPC-1 and 0.077 for FRPC-2) than that (0.0578) of the pure PC, which reveals that the DASO plays FR roles on gas phase which causing the occurrence of more serious incomplete combustion.

### 3.5. Characterization of Char Residue 

The digital and SEM images of char residues are present in Figure 4. As seen, the pristine PC leaves intumescent char residues with many holes on the surface, which provides the channel for the transfer of heat and mass. In comparison, a smooth and continuous surface, although a tiny crack, still exists for FRPC-1 composite. With the DASO content up to 2 wt%, comparatively, char residue gives an expansion height of ~8.4 cm and becomes more continuous and smoother, which keeps transmission of combustibles and heat from the external surface to the inner matrix [32].

As seen in Raman spectra of char residues (Appendix A), two sharp peaks appear at 1360 cm^−1^ (D band) and 1600 cm^−1^ (G band), which are assigned to the disordered carbon impurities (or defects) and 2D hexagonal graphitic lattice, respectively [33,34]. Normally, a lower area ratio (A_D_/A_G_) of the two peaks reveals a higher graphitization degree. It is found that the A_D_/A_G_ value is decreased from 2.92 for pure PC to 2.41 for FRPC-1 and 2.26 for FRPC-2, revealing that the DASO has a catalytic effect on the graphitization of the char residues.

Figure 5 shows the XPS survey spectra of pure PC and FRPC-2, and the resultant elemental content is listed in Table 4. It is obvious that FRPC-2 contains a small amount of P, Si, and N elements besides C and O elements in the pure PC, indicating that the three flame retardant elements play roles in the condensed phase. In the C1s spectra (Figure 5b), the binding energy peaks at 283.4~283.5 and 284.4~285.1 eV are assigned to C-C/C-H and C-O, respectively. In addition, a new peak at 287.2 eV belongs to C=O for FRPC-2. The O1s spectra (Figure 5c) show that the binding energy peaks located at 531.3~531.5 eV and 532.1~532.8 eV correspond to O-C and C=O, respectively. The peak intensity ratio of C=O/C-O is significantly reduced, indicating that substances with C=O basically overflow into the gas phase, while C-O (ethers and cross-linked structures) remain in the char residues [35]. Regarding the Si2p spectra (Figure 5d), the binding energy peak at ~101.0, ~102.0, and ~103.8 eV are attributed to the bonding states of Si-C, Si-O-C, and Si-O-Si, respectively. It indicates that DASO is thermally decomposed to produce silicon-containing fragments during combustion.

### 3.6. TGA-FTIR Analysis of Gaseous Products

The pyrolysis gases of neat PC and FRPCs at different temperatures were analyzed via TGA-FTIR spectra in order to understand the flame-retardant roles of DASO in the gas phase. As shown in Figure 6a–c, some common gases are produced for all specimens, including H_2_O (3658 cm^−1^), hydrocarbons (2800–3100 cm^−1^), CO_2_ (2300–2400 cm^−1^), CO (2080–2190 cm^−1^), phenyl compounds (1514 cm^−1^), aromatic ethers (1236 cm^−1^), and phenol derivatives (1174 cm^−1^). It is also found that two new peaks at 1258 cm^−1^ and 1068 cm^−1^ occur in FTIR spectra of FRPCs, which belong to P=O and P-O-C [36,37], respectively. The nonflammable gases (H_2_O and CO_2_) can act as diluting agents for oxygen and combustible volatiles and work synergistically with a small amount of phosphorus-containing fragments as radical scavengers to exert flame-retardant roles in the gas phase [38,39]. Additionally, the 3D real-time evolution TG-FTIR images (Figure 6a_1_–c_1_) of gaseous products show that the absorbance intensity tends to reduce visibly with the incorporation of DASO. Specifically, Appendix A shows the change of absorbance intensity of four typical flammable gases (CH_4_, phenyl, ether, and C-OH compounds) with time. As seen, the absorbance intensity of flammable volatiles decreases significantly during the decomposition of FRPC composites. It indicates that DASO can suppress the production of flammable volatiles, thus playing the flame retardant role in the gas phase.

### 3.7. Possible Flame Retardant Mechanism

Taking the described-above flame retardant role of DASO in both gas and condensed phase into consideration, a possible flame retardant mechanism is proposed and illustrated in Figure 2. With respect to the gas phase, nonflammable gases (H_2_O and CO_2_) dilute oxygen and flammable volatiles and work synergistically with phosphorus-containing radicals (·PO and ·PO_2_) as radical scavengers to capture the highly active radicals (H· and HO·) during the combustion process, further combustion is consequently slowed down. As to the condensed phase, the phosphorus- and silicone-containing fragments promote and participate in the formation of a dense and highly graphited char layer with Si-O-C and phosphorus-rich structure during the combustion, which acts as a physical barrier to inhibit the transfer of heat and flammable volatiles. 

### 3.8. Mechanical Properties

The influence of DASO on mechanical performances was evaluated according to tensile and impact testing, and the specific data are summarized in Table 5. By comparison with the pristine PC presenting tensile strength of 60.3 MPa, elongation at break of 87.6%, and impact strength of 68.9 kJ/m^2^, the FRPC composites keep almost the same elongation at break whilst their tensile and impact strength are reduced. For instance, the FRPC-2 presents tensile strength of 54.3 MPa and 57.2 kJ/m^2^, with a reduction of 10.0% and 17.0% compared to the pure PC, respectively. As we know, the mechanical performances of the composite correlate with the microstructure of the multiphase. As seen in Figure 7, all specimens show impact-fractured rough surfaces with ductile fracture characteristics. However, it is noted that FRPCs present obvious droplets, which act as stress concentration points, and the mechanical properties are reduced accordingly.

## 4. Conclusions

In summary, a novel Si/N/P-containing flame retardant (DASO) was synthesized through an Atherton–Todd reaction between DOPO and aminophenyl silicone oil, and its chemical structure was confirmed in terms of FTIR, ^1^H NMR, and ^31^P NMR. Furthermore, the DASO was adopted to improve the fire safety of PC. A V-0 rating and a high LOI of 32.2% were obtained for FRPC-2. Meanwhile, its PHRR and TSP were reduced by 26% and 44% by comparison with neat PC, respectively. The comprehensive analysis of char residues and pyrolysis gases showed that DASO played a synergistic flame retardant effect in both the gas and condensed phase. In the gas phase, it included the quenching effect of the free radicals (·PO and ·PO_2_) and the diluting effect of the nonflammable gases (H_2_O and CO_2_). Regarding the condensed phase, the phosphorus- and silicone-containing compounds produced from the degradation of DASO promoted and participated in the formation of a dense and highly graphited char layer, which acted as a physical barrier to inhibit the transfer of heat and flammable volatiles. Furthermore, the FRPCs keep almost the same elongation at break whilst their tensile and impact strength are reduced slightly.

## Data Availability

Not applicable.

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
