# Peer review of "A DOPO-Based Compound Containing Aminophenyl Silicone Oil for Reducing Fire Hazards of Polycarbonate"

_materials, 2023, doi:10.3390/ma16041449_

Round 1

Reviewer 1 Report

In this manuscript, the authors synthesized a P/N/Si-containing flame retardant named DASO) using the Atherton-Todd reaction between 9,10-dihydro-9-oxa-10-phospha-phenanthrene-10-oxide and aminophenyl silicone oil for reducing fire hazards of polycarbonate. The chemical structure of DASO was well characterized  via FTIR, 1H and 31P NMR. The flame retardant effect obtained is significant and the manuscript was well written in my opinion and should be published in Materials. However, I suggest the authors do the following changes to help improve the manuscript further.

Comments

1. The novelty /research gap this work intends to fill has not be clearly stated from the literature review under the introduction section. 

2. For the sake of repeatability, the authors can state all the chemicals and materials used for the experiment indicating their level of purity and where they were purchased. 

3. A few grammatical and typographical errors were noticed throughout the manuscript. Pls read through carefully and correct them. Eg. To a 250 mL three-neck flask equipped with a magnetic stirrer and a reflux........., "conten", "As we all Known", and many more.

4. Results of the 31P NMR is missing. In any case, the position of phosphorus shouldn't be too affected despite the chemical environment. Kindly provide the result of the 31 P NMR. 

5. What is the yield of DASO from the synthesis and how pure is the product?

6. From the flame retardant mechanism proposed, a lot is mentioned about radicals, are these radicals confirmed from this study they only imaginary considering the fact that radicals have a short transient stage and TG-IR cannot capture them. Formation of the radicals mentioned in this work has been confirmed in other studies, kindly reference accordingly.

Author Response

Q1. The novelty /research gap this work intends to fill has not be clearly stated from the literature review under the introduction section. 

Reply: Thanks for your valuable comment. To emphasize the novelty of the present work, the following sentences were added in the revised version. However, the introduction of additive FRs sometimes deteriorates the mechanical properties of the PC composite due to their poor compatibility, especially in the case of the usage of anti-dripping agent such as talc and PTFE. Therefore, it is still a challenge to endow PC matrix with better flame retardancy meanwhile keep almost the mechanical properties. The effect stratagem is to improve the compatibility of the FRs with PC or avoid the usage of anti-dripping agent.  As expected, UL-94 V0 rating was obtained for PC/2 wt% DASO composite accompanying with comparable mechanical properties.

Q2. For the sake of repeatability, the authors can state all the chemicals and materials used for the experiment indicating their level of purity and where they were purchased. 

Reply: We thank the reviewer for the valuable suggestion. These descriptions were added to the supplementary document.

Q3. A few grammatical and typographical errors were noticed throughout the manuscript. Pls read through carefully and correct them. Eg. To a 250 mL three-neck flask equipped with a magnetic stirrer and a reflux........., "conten", "As we all Known", and many more.

Reply: We thank the reviewer for the valuable suggestion, and checked grammatical and typographical errors carefully.

Q4. Results of the 31P NMR is missing. In any case, the position of phosphorus shouldn't be too affected despite the chemical environment. Kindly provide the result of the 31 P NMR. 

Reply: Thank the reviewer for the valuable suggestion, and the results of 31P NMR spectra were discussed in Section 3.1.

Q5. What is the yield of DASO from the synthesis and how pure is the product?

Reply: We sincerely thank the reviewer for valuable comments to improve the manuscript. The yield of DASO is about 91% and high purity of the product can be characterized by nuclear magnetic test.

Q6. From the flame retardant mechanism proposed, a lot is mentioned about radicals, are these radicals confirmed from this study they only imaginary considering the fact that radicals have a short transient stage and TG-IR cannot capture them. Formation of the radicals mentioned in this work has been confirmed in other studies, kindly reference accordingly.

Reply: We sincerely thank the reviewer for valuable comments to improve the manuscript, and two references were added in the revised manuscript.

Reviewer 2 Report

I want to congratulate the authors for a very interesting and clear article. I read it with the pleasure.

1.      In this manuscript, the authors synthesized a P/N/Si-containing flame retardant called DASO) using the Atherton-Todd reaction between 9,10-dihydro-9-oxa-10-phospha-phenanthrene-10-oxide and aminophenyl silicone oil in to reduce the fire hazard of polycarbonate. A significant flame retardant effect was obtained. The authors also performed a full characterization of the obtained compounds and materials.

2.      I believe that the article is original and will complement the literature available so far with possible additional flame retardancy of polycarbonate.

3.      Thanks to the extensive analysis of both thermal and mechanical, as well as flammability and smoke emission, the presented solution was comprehensively described. From the knowledge available to me, I can say that this article supplements the available literature with new solutions.

4.      A hint for the authors for the future would be to analyze the combustion of the produced material in fire conditions and to analyze the substances present in the generated smoke. TGA/FT-IR analysis is very popular, one is that very small samples are used to perform it. A method worth considering is, for example, a Purser furnace combined with a GC-MS or with a spectrometer calibrated to the various gases available.

5.      I have no objections to the conclusions.

6.      The number and selection of literature items does not raise any objections.

7.      The charts, photos and tables are legible and do not require correction.

Small Notes:

 - Please add a subsection about the method of conducting individual research methods is missing.

- Please add two or three sentences about novelty in this work, to emphasize it.

Author Response

- Please add a subsection about the method of conducting individual research methods is missing.

Reply: We sincerely thank the reviewer for valuable comments to improve the manuscript. The subsection about the method was added in the Supporting information.

- Please add two or three sentences about novelty in this work, to emphasize it.

Reply: Thanks for your valuable comment. To emphasize the novelty of the present work, the following sentences were added in the revised version. However, the introduction of additive FRs sometimes deteriorates the mechanical properties of the PC composite due to their poor compatibility, especially in the case of the usage of anti-dripping agent such as talc and PTFE. Therefore, it is still a challenge to endow PC matrix with better flame retardancy meanwhile keep almost the mechanical properties. The effect stratagem is to improve the compatibility of the FRs with PC or avoid the usage of anti-dripping agent.  As expected, UL-94 V0 rating was obtained for PC/2 wt% DASO composite accompanying with comparable mechanical properties.